# The Rab Geranylgeranyl Transferase Beta Subunit Is Essential for Embryo and Seed Development in *Arabidopsis thaliana*

**DOI:** 10.3390/ijms22157907

**Published:** 2021-07-24

**Authors:** Joanna Rojek, Matthew R. Tucker, Michał Rychłowski, Julita Nowakowska, Małgorzata Gutkowska

**Affiliations:** 1Faculty of Biology, University of Gdańsk, 80-308 Gdańsk, Poland; joanna.rojek@ug.edu.pl; 2Waite Research Institute, School of Agriculture, Food and Wine, The University of Adelaide, Urrbrae, SA 5064, Australia; matthew.tucker@adelaide.edu.au; 3Intercollegiate Faculty of Biotechnology, University of Gdańsk, 80-307 Gdańsk, Poland; michal.rychlowski@biotech.ug.edu.pl; 4Faculty of Biology, University of Warsaw, 02-096 Warsaw, Poland; julita@biol.uw.edu.pl; 5Institute of Biochemistry and Biophysics, Polish Academy of Sciences, 02-106 Warsaw, Poland

**Keywords:** *Arabidopsis*, auxin, embryo, endosperm, PIN1, Rab, Rab Geranylgeranyl Transferase, seed, seed coat

## Abstract

Auxin is a key regulator of plant development affecting the formation and maturation of reproductive structures. The apoplastic route of auxin transport engages influx and efflux facilitators from the PIN, AUX and ABCB families. The polar localization of these proteins and constant recycling from the plasma membrane to endosomes is dependent on Rab-mediated vesicular traffic. Rab proteins are anchored to membranes via posttranslational addition of two geranylgeranyl moieties by the Rab Geranylgeranyl Transferase enzyme (RGT), which consists of RGTA, RGTB and REP subunits. Here, we present data showing that seed development in the *rgtb1* mutant, with decreased vesicular transport capacity, is disturbed. Both pre- and post-fertilization events are affected, leading to a decrease in seed yield. Pollen tube recognition at the stigma and its guidance to the micropyle is compromised and the seed coat forms incorrectly. Excess auxin in the sporophytic tissues of the ovule in the *rgtb1* plants leads to an increased tendency of autonomous endosperm formation in unfertilized ovules and influences embryo development in a maternal sporophytic manner. The results show the importance of vesicular traffic for sexual reproduction in flowering plants, and highlight RGTB1 as a key component of sporophytic-filial signaling.

## 1. Introduction

Flowering plants, which dominate present-day flora, have developed a sophisticated and successful system of propagation by seeds. Unlike other plants, the angiosperm female gametophyte (FG) undergoes double fertilization. The haploid egg cell is fertilized by one sperm cell to form a zygote [1]. Mitotic divisions of the zygote follow and give rise to the embryo, which by the 16-cell globular stage exhibits differentiated inner and outer cell layers. In *Arabidopsis*, when the embryo contains 32 cells, it undergoes a transition from globular to bilateral symmetry, initiating cotyledons (embryonic leaves) and a radicle (embryonic root). The endosperm arises from the diploid central cell, which is fertilized by the second sperm cell [1]. The initial divisions of the triploid nuclei in the endosperm are not followed by a cytokinesis and the endosperm, therefore, develops as a syncytium; only when the embryo has reached early heart stage does the endosperm cellularize [2]. In some plants an adaptive mechanism enables formation of the embryo independently from fertilization, also referred to as apomixis [3,4]. Autonomous endosperm (AE) development is also triggered under stress in vivo/in vitro or in a number of mutants [5,6,7]. The outermost component of the seed is the seed coat, which forms entirely from maternal tissues. Its precursors, the integuments, surround and mechanically protect the developing female gametophyte. Moreover, they supply the FG with nutrients and hormones [8] and fulfil an important role in fertilization as a source of chemical attractants for pollen tube growth and guidance [9,10]. The seed coat protects the developing embryo from mechanical stress, UV radiation and water loss. It plays a role in dissemination and later in dormancy maintenance in the mature seed. Finally, during seed germination, the seed coat exudes polysaccharides that form a safe microenvironment for radicle growth [11,12].

Auxin is a hormonal master regulator of plant development and growth. In particular, it controls the development of pre- and post-fertilization reproductive structures in plants (reviewed in [13]). In angiosperms, auxin promotes formation of ovules in the ovary [14], female gametophyte development [15,16,17], pollen development [18,19,20] and the synchronization of anther and ovule maturation [8]. Auxin is also responsible for growth and differentiation of the embryo and the endosperm after double fertilization [21]. Simultaneously auxin influences growth of the seed coat [22]. Auxin is a factor that enables communication of the three main seed tissues: the embryo, the endosperm and the seed coat [23,24,25,26]. The source of auxin for pre-anthesis, fertilization and post-fertilization stages and early embryo development is local synthesis in the sporophytic tissues of the ovary, in particular in integuments surrounding the female gametophyte [16,27,28]. Its transport to the vicinity of the embryo sac is achieved by the means of polar efflux by the PIN family of transporters localized in the funiculus and the integuments [8,15] and auxin importers from the AUX and ABCB families [29]. Auxin is thought to be absent from the embryo sac throughout its development [8,16], although some controversy remains [27]. Later, after the egg cell fertilization, the auxin biosynthesis genes are expressed at the embryo suspensor [30,31], and finally, starting from 16-cell globular stage, the embryo becomes self-sufficient for auxin biosynthesis [29,31,32,33]. Around fertilization, the main PIN proteins responsible for auxin homeostasis in the ovule are PIN1 and PIN3 [8]. PIN1 is expressed in the embryo proper from the four-cell stage, and later in the epidermis of the cotyledon precursors followed by the pre-vascular strands in both the cotyledons and hypocotyl [34,35,36]. PIN2 and PIN3 are not expressed in the embryo or seed structures until the late stages, but PIN4 and PIN7 play an important role in the apical-basal axis establishment of the embryo [37,38]. In particular, the diversion of PIN7 localization in the suspensor from apical to basal membranes is coincident with the major step in embryo development: the change of symmetry from globular to bilateral [33]. PIN4 also enables auxin to accumulate at the prospective root meristem [39,40]. 

PIN proteins are transported in a polar fashion to their destination, the plasma membrane, by means of vesicular transport; the regulation of their activity in the cell is also achieved by a constant recycling from membranes to internal vesicles and back [41,42]. Among proteins involved in this membrane recycling are small GTPases from the Rab family [43,44,45]. Rab proteins are key components of vesicular traffic machinery in all Eukaryotes [46,47]. They “label” vesicles to their final destination, enable their movement via the cytoskeleton and mediate complex formation between membrane fusion machinery proteins located on the donor vesicle and the acceptor membrane [46,48]. Small GTPases are post-translationally modified to enable membrane anchoring [48]. The lipid modification characteristic for Rab proteins is geranylgeranylation [49], which provides the proteins with affinity for membranes, comparable to modification with a long unsaturated fatty acid [50]. Introducing such hydrophobic moieties on two neighboring cysteine residues in a Rab polypeptide by a stable tio-ether bond renders the protein membrane-bound [51]. This lipid modification is crucial for Rab cellular function as a vesicle fusion and fission facilitator [46].

Rab proteins are geranylgeranylated by an enzymatic complex Rab Geranylgeranyl Transferase (RGT), which consists of a tightly bound lipid binding RGTB subunit, a catalytic RGTA subunit and an auxiliary Rab-binding REP (Rab Escort Protein) subunit [52]. RGTB activity is encoded in *Arabidopsis* by two genes: *RGTB1* and *RGTB2* [53]. In previous studies, the role of RGTB proteins in the development of the *Arabidopsis* sporophyte [53,54] and both male and female gametophytes [15,54] was described, but the embryonic development in *rgtb1* mutants has not been studied. In the mature sporophytic tissues, RGTB1 is the more abundant protein isoform and knock-out mutations cause pleiotropic phenotypes in all *rgtb1* plants [53]. RGTB2 protein is less abundant in the sporophytic tissues and a knock-out of its gene causes only minor defects in polar growth; moreover, the effect of this mutation is not fully penetrant [54]. Both genes are expressed in pollen and are responsible for pollen development and pollen tube growth; the combined knock-out of *RGTB1* and *RGTB2* causes total male sterility [54]. Recently, we described the influence of the *rgtb1* mutation on female gametophyte formation [15]. Disturbance of the Rab-mediated vesicle recycling of PIN1 and PIN3 auxin transporters in the funiculus caused auxin accumulation in the ovule and impeded progression of female gametophyte development in many ovules [15]. We also noticed some irregularities in integument formation. Here, we further explore the role of Rab geranylgeranylation during seed and embryo formation to understand how vesicular traffic influences plant reproduction.

## 2. Results

### 2.1. rgtb1 Mutants Form Fewer Seeds than WT Plants

The number of ovules in the mature pistils of the *rtgb1*/*rgtb1* plants was approximately half that of WT [15]. Moreover, a significant fraction of around 20–25% of female gametophytes were arrested at the FG1/FG2 stages of development, which coincided with PIN1 transporter mislocalization and auxin accumulation in the ovules [15]. However, when the *rgtb1*/*rgtb1* flowers were hand-pollinated by either *RGTB1* or the *rgtb1* pollen, seeds were recovered, but always in amounts lower than for WT flowers (we will use the name WT meaning the *RGTB1*/*RGTB1* genotype throughout the entire manuscript*)* (Figure 1).

The failure of seed formation in the case of open pollination of the *rgtb1*/*rgtb1* plant may be due to the discrepancy in anther/pistil size [53], while poor success of hand pollination of the *rgtb1*/*rgtb1* may be explained by low fertility of the *rgtb1*/*rgtb1*-derived pollen [54]. Interestingly, the amount of seed formation in the *rgtb1*/*rgtb1 ×* WT manual cross, which on average was less than 20% (compared to the WT × WT hand cross; Figure 1A), is much lower than expected from the ovule maturation data (a 50% reduction in ovules, of which 20–25% are non-functional, would predict a ~40% ratio of a successful seed formation [15]). Therefore, we decided to examine the processes of pollination, fertilization and seed formation in *rgtb1*/*rgtb1* mutants.

### 2.2. Pre-Fertilization Events May Account for Failure of Seed Formation in rgtb1 Mutants

Because the reduced capacity of vesicular transport in the *rgtb1*/*rgtb1* plants might affect pollen behavior around the time of pollination, we checked pollen receptivity at the stigmatic papillae by scoring its recognition and hydration. Stigmatic papillae in the *rgtb1*/*rgtb1* mutant developed normally (Appendix A). Inefficient recognition of pollen on the stigma results in pollen grain detachment or degeneration [55]. Interestingly, WT pollen grains hydrated and germinated on *rgtb1*/*rgtb1* stigmas upon hand pollination but with 30% lower efficiency than on WT stigma (Figure 2A,E,I,M).

WT pollen tubes managed to grow through the transmission tract of the style in the *rgtb1*/*rgtb1* mutants (Figure 2B,F,J), but more pollen tubes were mistargeted. Approximately 65% of the WT tubes found the micropyle in *rgtb1*/*rgtb1* compared to 100% WT pollen tubes in WT flowers (Figure 2C,D). Additionally, 35% of WT pollen tubes showed abnormal behavior in the vicinity of the micropyle in *rgtb1*/*rgtb1* flowers (Figure 2N); tubes meandered randomly (Figure 2G,K,O), more than one tube approached the micropyle (Figure 2G,K) or the pollen tubes missed the micropyle of the unfertilized ovule (Figure 2O). Because all of the pollen used for this experiment was WT, the atypical behavior of the pollen tubes clearly depended on the ovule genotype. The ovules that attracted pollen tubes and enabled their entry exhibited a normal FG, while those showing difficulties in pollen tube guidance or entry lacked a female gametophyte or showed an arrested one, even though their integuments had developed correctly. Oftentimes, they showed an abnormal callose plug in the place of the egg apparatus (Figure 2H,L).

Deficiencies in the pre-fertilization events may account for the decrease in seed formation efficiency in the *rgtb1*/*rgtb1*. However, we were also interested in the fate of the apparently normal, mature-looking *rgtb1*/*rgtb1* ovules that showed pollen tube entry.

### 2.3. Autonomous Endosperm Formation Is More Frequent in rgtb1 Than in WT Plants

*Arabidopsis* is a self-fertilizing plant, and in WT flowers, the opening of the flower is simultaneous with self-pollination. In contrast, in the *rgtb1*/*rgtb1* mutant, self-pollination is a very rare event. Mutant *rgtb1*/*rgtb1* pistils typically showed unfertilized mature ovules, shrunken aborted ovules without FGs, and apparently normal ovules containing only endosperm (Figure 3A–C,F,G). Young embryos were observed in very few cases (Figure 3E). In contrast, in WT flowers, open pollination nearly always led to correct development of the post-fertilization structures, the embryo and the endosperm (Figure 3C,D), and extremely rarely to endosperm only. 

From the point of view of self-pollination capacity, *rgtb1*/*rgtb1* flowers were (nearly) equivalent to emasculated WT flowers. Indeed, the *rgtb1*/*rgtb1* anthers were smaller (Appendix A) [15,53,54], delayed in tapetum maturation (Appendix A) and indehiscent (Appendix A), and therefore, unable to correctly release pollen (Appendix A). After manual emasculation of flower buds at the pre-anthesis stage, in WT plants, most gametophytes paused at FG maturity. In 2.8% of viable unfertilized ovules, the central cell began to divide giving rise to autonomous endosperm, consistent with previous reports [26] (Figure 3H,I). Interestingly, the percentage of viable, non-degenerated ovules forming autonomous endosperm was always high in the *rgtb1*/*rgtb1* flowers, reaching 10% (Figure 3H,J), independent of emasculation.

Autonomous endosperm formation requires high auxin levels in its precursor, the central cell, but the presence of auxin in the central cell was barely detected in previous studies [26]. In our experiments, in emasculated WT flowers, the central cell nucleus often showed *pDR5rev:*3xVenus-NLS reporter activity [56], revealing a detectable auxin response in the central cell before fertilization (Figure 4A and Figure 5A). In cases where further divisions of the unfertilized central cell nucleus followed, the autonomous endosperm nuclei in WT plants also showed the *pDR5rev:*3xVenus-NLS auxin-dependent signal (Figure 5B). The signal arising from the DIIS-Venus reporter [57] was detected in WT in the sporophytic tissues of the ovule only after emasculation (Figure 5C), marking auxin absence in the integuments and chalaza, while strong *pDR5rev*:3xVenus-NLS signal dominated in the same structures after fertilization (Figure 4B–D), marking auxin presence in the integuments and chalaza.

Autonomous endosperm in *rgtb1*/*rgtb1* ovules from emasculated flowers initiated with a similar frequency compared to non-emasculated flowers, but was higher than that in the WT flowers (Figure 3H). We hypothesized that this may relate to high auxin levels in the *rgtb1* central cell, possibly as a result of lower auxin efflux from the ovule during FG development, as was shown in [15]. This would also be consistent with the observation that treatment of the flowers with exogenous auxin analogues (in particular 2,4-D) induces autonomous endosperm formation in a large portion of WT ovules [26]. Surprisingly, the *pDR5rev:*3xVenus-NLS auxin reporter was not detected, or detected at a very low level, in the central cell nucleus or autonomous endosperm nuclei of *rgtb1*/*rgtb1* ovules (Figure 4E,F and Figure 5D,E,G,H), in contrast to the situation in WT flowers (Figure 5A,B). The opposite situation was observed with the DIIS-Venus reporter, which was present in the integuments of emasculated WT flowers, showing that auxin levels are low (Figure 5C), but was lacking in the integuments of the *rgtb1*/*rgtb1* non-fertilized ovules, suggesting the presence of auxin inside them (Figure 5F,I).

### 2.4. Large Malformed Embryos Form in Seeds of rgtb1 Plant

A significant fraction of the *rgtb1*/*rgtb1* ovules were successfully fertilized, despite the problems described above. We followed embryo development in the WT, the *RGTB1*/*rgtb1* self-cross and the *rgtb1*/*rgtb1 ×* WT siliques (Figure 6 and Appendix A). While the WT embryos were uniform in size and developmental stage in the siliques analyzed on the same day after pollination, the *rgtb1*/*rgtb1 ×* WT embryos were not synchronized, resembling the situation that was observed in ovule development in the same ovary in the *rgtb1*/*rgtb1* plants [15]. We found major differences between the genotypes starting from the transition/early heart stage of embryogenesis (Figure 6). In *RGTB1*/*rgtb1* plants, which produce WT:*RGTB1*/*rgtb1*:*rgtb1*/*rgtb1* progeny in a 1:2:1 ratio, only 2.8% of embryos were wider, larger or otherwise deformed (Figure 6I–L and Appendix A). After pollination of *rgtb1*/*rgtb1* pistils with WT pollen only, the *RGTB1*/*rgtb1* embryos were expected. To our surprise we found that more than 10% of the *RGTB1*/*rgtb1* embryos developing on the *rgtb1*/*rgtb1* plants pollinated with WT were larger than their WT counterparts (WT *×* WT cross; Figure 6E–H compare with Figure 6A–D, quantification shown on Appendix A). Their body plan was more robust, with wider and shorter cotyledons and wider hypocotyl at the base of the cotyledons, while no changes in the hypophysis and other basal parts of the embryo were noticed (Figure 6E–H). These embryo deformations were not accompanied by major growth abnormalities, such as cup-shaped fused cotyledons or multiple cotyledons. Asymmetry in the cotyledon length was quite common (Figure 6H). The same observation of the embryo shape was made for the *rgtb1*/*rgtb1* cross with the *RGTB1*/*rgtb1*-derived pollen (Figure 6M–P). In this case, the ratio of the genotypes was *RGTB1*/*rgtb1:rgtb1*/*rgtb1* 1:1; however, far less than half of the embryos were deformed (Appendix A), supporting the notion that it is not the genotype of the embryo itself that is responsible for the observed phenotype.

Deformations in embryos usually lead to germination defects or seedling lethality. Seedlings and mature *RGTB1*/*rgtb1* plants from the *RGTB1*/*rgtb1* progeny were indistinguishable from WT [54]. The number of dwarf seedlings or non-germinating seeds in the *RGTB1*/*rgtb1* self-cross was not significantly higher than the 1/4 expected from the Mendelian segregation (28% out of 212 in case of the *rgtb1*-*1* and 25% out of 220 in case of the *rgtb1*-*2* mutants; in this case, the pool of the dwarf seedlings and non-germinating seeds represent the *rgtb1*/*rgtb1* progeny, Appendix A). In contrast, we found 12% dwarf seedling and non-germinated seeds in the progeny of *rgtb1*/*rgtb1 ×* WT (137 were counted, all of *RGTB1*/*rgtb1* genotype, Appendix A) but no such seedlings in WT *×* WT cross (182 were counted, Appendix A). 

### 2.5. Deformations in Embryo Shape in rgtb1 × WT Crosses Correlate with Lower and Diffused Auxin Maxima at the Cotyledon Tips

Because cotyledon outgrowth is a process dependent on auxin maxima formation [38] we decided to trace the auxin distribution during embryo growth in the *rgtb1*/*rgtb1* versus *RGTB1*/*rgtb1* plants (Figure 7). During the early globular stage, *pDR5rev:*3xVenus-NLS activity was strong at the boundary of the embryo proper (Figure 7A). At the early heart to late heart stage, *pDR5rev:*3xVenus-NLS was detected in the basal cells and at the cotyledon primordia (Figure 7B–D), and at the torpedo and upright stage was detected in the cotyledon veins, radicle vasculature and the developing shoot apical meristem (Figure 7E). This held true for the WT *×* WT cross progeny as well as for the *RGTB1*/*rgtb1* self-cross progeny. Strikingly, in 10% of the *rgtb1*/*rgtb1 ×* WT embryos, starting from the heart stage, the auxin maxima in the tips of cotyledons were weaker in intensity and incorporated a lower number of cells (Figure 7H–K). These particular embryos often showed cotyledon shortening and widening or asymmetry.

### 2.6. PIN1 Efflux Protein Is Mislocalized in the Deformed rgtb1 × WT Embryos

PIN1 is the main auxin transporter involved in auxin localization in the apical half of the *Arabidopsis* embryo [58]; hence, we studied its distribution (Figure 8). Starting from the late globular stage, the PIN1 distribution differed in WT and many *rgtb1*/*rgtb1 ×* WT embryos. In WT, PIN1 was localized mainly on cell membranes in a polar fashion (Figure 8A–C), presumably mediating auxin transport to the prospective tips of the cotyledons (recently reviewed in [59]). At the early heart stage, when the cotyledon primordia were visible, there were two discernible PIN1 expression domains at the embryo surface (Figure 8B) [58]. The site of auxin biosynthesis between them was devoid of PIN1 expression [59]. On the contrary, in many *rgtb1*/*rgtb1 ×* WT embryos, the PIN1 domain remained wider, also covering the region of the embryo between the emerging cotyledon tips (Figure 8G,K,L,N). In *rgtb1*/*rgtb1 ×* WT embryos, PIN1 was often internalized in vesicles inside the cells (Figure 8H,J–O). At later stages of development, another auxin-dependent process was delayed in many *rgtb1*/*rgtb1 ×* WT embryos—the formation of the embryo vasculature [60] (Figure 8I,J,O vs. Figure 8C–E).

The distribution and localization of two other auxin efflux transporters, PIN4 and PIN7, is also important for embryo development in *Arabidopsis* [37,38,39,40]. They are crucial for the formation of the basal parts of the embryo, the hypophysis and the prospective root apical meristem. However, in *rgtb1*/*rgtb1 ×* WT embryos, we never observed additional vertical divisions of cells in the suspensor region or the lens-shaped hypophysis precursor cell that are indicative of PIN4 or PIN7 absence or mislocalization [33,37,38,39,40] (Figure 6, Figure 7 and Figure 8).

### 2.7. Seed Coat Morphology in rgtb1 Mutants Is Defective

Pollination of the *rgtb1*/*rgtb1* plants with WT pollen gave a large fraction of seeds that progressed to maturity (Appendix A). The mature seeds of the *rgtb1*/*rgtb1* plants pollinated with WT pollen germinated less efficiently than the WT control seeds (91.25% germinated versus 100% germination in WT, Appendix A). Moreover, some differences in the seed coat appearance were obvious when WT and *rgtb1*/*rgtb1 ×* WT seeds were compared (Figure 9A–E). WT seeds exhibit hexagonal epidermal cells with a volcano-shaped structure (the columella) in the center of each cell (Figure 9A). The *rgtb1*/*rgtb1*-derived seeds had irregularly shaped epidermal cells with cell walls thinner than in the WT; the volcano-shaped mucilage pockets were absent (Figure 9D,E). This trait was previously described to coincide with an inability to secrete oligosaccharides during the hydration step of germination in transport-related mutants [61,62]. In the *rgtb1*/*rgtb1-*derived seeds, the mucilage layer formed during seed imbibition was thinner and more intensely stained than in WT seeds, which formed a thicker and more cloudy mucilage sheath (Figure 9F–J). Oftentimes, the mucilage layer in the *rgtb1*/*rgtb1-*derived seeds was unevenly distributed (Figure 9G). Neither the seed coat surface nor mucilage formation was changed in seeds formed on the *RGTB1*/*rgtb1* plants (Figure 9B,C) in comparison to the WT (Figure 9A,F,H).

### 2.8. Expression of the RGT-Encoding Genes Change during Embryo Development

To gain insight into the Rab geranylgeranylation processes during *rgtb1* embryo development, we performed a detailed in silico analysis of the expression of the RGT complex encoding genes based on published data [63] (Figure 10). There are two RGTB encoding genes in *Arabidopsis*, of which *RGTB1* is abundantly and ubiquitously expressed in the sporophyte and *RGTB2* is about 10–20 times less abundant in the sporophyte [53]. Both genes are expressed to a comparable or even higher level in the gametophytes, with *RGTB2* dominating [15,53]. In the embryo, both genes were expressed, with the *RGTB2* showing higher expression at the early stages and *RGTB1* showing higher expression starting from globular until late heart stages (Figure 10). Interestingly the other genes coding for the RGT complex components were also transcribed in a developmentally regulated fashion. In particular, the expression of the *REP* gene was high at the globular to heart stages, resembling *RGTB1* (Figure 10). The expression of the *RGTA1* gene expression peaked at the very early stages (as *RGTB2*), then decreased and again increased at the early heart stage (as *REP* and *RGTB1*). The expression of *RGTA2* was high only at the earliest stages of embryogenesis, but it is uncertain if this gene encodes for a functional protein or is just a pseudogene [64]. 

Rab-encoding genes may be grouped based on their stage of expression during embryogenesis, such as early (for example *Rab A3*, *Rab D2b*, *Rab A4a*, *Rab H1a*, *Rab A1h*, *Rab A4d* and *Rab G2*), mid (for example *Rab E1e*, *Rab A2d*, *Rab G3c*, *Rab G3a*, *Rab A1c* and *Rab A2a*), late (for example *Rab G3f*, *Rab C2a*, *Rab A5d*, *Rab B1c* and *Rab D2c*) and ubiquitous (for example *Rab A1g* and *Rab G1*) (Figure 10). This suggests that the transcription of different genes encoding members of the same Rab family are differentially regulated, but for each developmental stage, the set of basic Rab-dependent function is present. Because the *rgtb1*/*rgtb1* sporophyte seems to influence embryo development, we also studied the transcription of RGT-encoding genes in the seed sporophytic tissues (based on data from [65]) and found that all of them were ubiquitously transcribed (Appendix A). Interestingly *RGTB1* is strongly transcribed in the seed coat at the very early stages of seed development, corresponding to the pre-globular phase of embryogenesis.

## 3. Discussion

### 3.1. Mislocalized and Weak Auxin Maxima Inhibit the Development of Multiple Reproductive Structures in rgtb1 Mutants

In a previous study, we examined female gametophyte development in the vesicular transport-deficient mutant *rgtb1* [15]. Abnormal recycling of the PIN1 and PIN3 auxin transporters in the funiculus and other sporophytic structures of the ovule led to auxin accumulation in the direct vicinity of the developing post-meiotic cells in a significant fraction of the ovules. This led to the female gametophyte arrest in these ovules as early as the first or second mitosis. The process appeared random and suggested that exceeding a threshold limit of auxin in certain ovules led to developmental arrest of the FG, while auxin levels just below the threshold did not induce such an effect [15]. A similar mode of auxin responsiveness was described earlier for stomata guard cell differentiation from meristemoid cells in the *pin3* mutant [66]. In this case, the accumulation of auxin in stomata precursors, via *PIN3* knockout, disturbed the equal division of these cells and formation of mature stomata. High auxin levels as a result of decreased efflux were also shown for another developmental process engaging asymmetric division: the mitosis of the unicellular microspore [20]. In both of these cases the small difference in auxin concentration in a certain structure leads to strikingly different effects of a binary nature. Interestingly, in many *rgtb1*/*rgtb1 ×* WT embryos, the arrest or delay in the development clearly seen for the early stages can be recovered at later, slower stages of development. This means that the embryos/seeds reach maturity in a more or less synchronized manner. This difference in the zero-to-one effects of auxin in the differentiation of megaspores, microspores and guard cells in comparison to the dose-dependent delay in embryos in the *rgtb1*/*rgtb1* plants may be explained by the multicellularity of the embryos [67,68,69]. In all the described structures, the increased concentration of auxin in the (sporophytic) tissues surrounding the cell/embryo seems to be inhibitory for realization of a correct developmental plan [70,71].

### 3.2. Increased Auxin Levels but Not Auxin-Dependent Transcription in the Central Cell Triggers Autonomous Endosperm Formation in the rgtb1 Mutant

The results concerning the nature of the phenotypes induced by deficiencies in Rab-dependent vesicular traffic in the *rgtb1* mutants [15,53,54] point to a generalized deregulation of auxin-dependent pathways. One interesting example of a structure that is formed in response to an increase in auxin level is the seed endosperm [26]. After fertilization (or external auxin administration), the auxin level in the central cell increases and the nucleus starts dividing [26]. Without fertilization, the auxin levels in the central cell are low and the mitotic divisions are suspended [26]. In the *rgtb1*/*rgtb1* mutant, the endosperm forms without fertilization in over 10% viable ovules, but surprisingly, no signal from the sensitive positive auxin reporter *pDR5rev*:3xVenus-NLS [56] was observed in the central cell or the endosperm nuclei. Moreover, the signal from the DIIS-Venus negative auxin reporter [57] is detected in the integuments, meaning that high auxin concentration in the integuments does not translate to an auxin transcriptional response in the embryo sac. In contrast, in WT unfertilized ovules, a strong response from the *pDR5rev*:3xVenus-NLS reporter was observed in the central cell, but the endosperm nucleus divisions rarely followed. 

This discrepancy in the auxin reporter behavior in the unfertilized central cell nucleus and its derivative, the endosperm, may be explained by the nature of reporters used in this study: DIIS-Venus is a reporter of auxin input signaling—a protein that is quickly degraded upon auxin binding [57]. The lack of the signal coming from DIIS reporter in the *rgtb1*/*rgtb1* integuments may reveal the presence of auxin in the sporophytic ovule tissues around maturity. *p**DR5rev:*3xVenus-NLS is an auxin output reporter, showing auxin-dependent transcriptional activity [56]. It is possible, therefore, that in the *rgtb1*/*rgtb1* ovules, auxin is present and leads to AE nuclei division, but the transcription of the auxin-dependent genes (having classical, consensus auxin response elements in their promoters) is compromised; hence, no signal from *pDR5rev* reporter is observed. The explanation may be that some fast process engaging auxin directly [72], not through auxin-dependent transcription, plays a role. One example of such a process is auxin-dependent plasmodesmata gating by callose [73,74], while another is the auxin-dependent pathway of plasma membrane depolarization and membrane protein degradation (reviewed in [75]). Of particular note is that in both explanations, the vesicular transport downturn seen in the *rgtb1* mutant may be directly detrimental: the transport of callose synthase is Rab-mediated [76] as well as the transport of the ubiquitinated membrane proteins for degradation (reviewed in [77]). Alternatively, a non-canonical transcription process may be engaged. The *pDR5rev* reporter is based exactly on a consensus auxin responsive element sequence binding IAA domain containing transcription regulators [78]. In contrast, the ETTIN (ARF3) auxin-dependent transcription factor takes part in many developmental processes, but does not bind the canonical ARE sequences [79]. Many of the phenotypes of the *ett* mutant, especially in ovule and seed formation [79], are similar to those exerted by the *rgtb1* mutation, for example an inner integument deformation, a lower seed set and excessive branching.

What is the source of the auxin in the central cell and the endosperm in the *rgtb1*/*rgtb1* mutants? It may originate from the remaining high auxin levels in the sporophytic tissues of the developing ovule. This abnormal auxin maximum is a result of disturbed PIN1 and PIN3 transport and/or recycling in the funiculus at earlier stages of FG development, leading to compromised auxin efflux out of the ovule [15]. One possibility is that some transporters from the PIN, AUX and ABCB families enable auxin influx into the embryo sac from the surrounding integument tissues at maturity [29,31]. A recent report also suggests that auxin can move from cell to cell through plasmodesmata, and that this process in some tissues may significantly contribute to auxin fluxes [80]. 

### 3.3. Deficiency in Vesicular Transport in the Sporophyte Affects Embryo Size and Shape in the rgtb1 Mutant

Despite the presumably high auxin level in the mature ovules of the *rgtb1*/*rgtb1* plants, the initial stages of embryo formation are not affected. This coincides with apparently normal PIN1-GFP and auxin (detected through *pDR5rev:*3xVenus-NLS reporter) distribution. The PIN4 and PIN7 auxin carriers are engaged in the early embryo development as well [31], but were not studied here. No major changes in cell division patterns, neither in the upper half of the embryo (PIN1-dependent [34]) nor in the lower half (PIN4-dependent [39]) or the suspensor (PIN7-dependent [38]) were noticed in embryos developing on the *rgtb1*/*rgtb1* plants before the late globular stage. Interestingly, 10% of embryos arising from the *rgtb1*/*rgtb1 ×* WT cross were larger and had shorter and/or asymmetric cotyledons. This developmental defect should be of sporophytic maternal nature, because the genotype of these embryos was *RGTB1*/*rgtb1*, and only 2.8% embryos that formed on the *RGTB1*/*rgtb1* plant showed such deformations. If this deformation was dependent on the genotype of the embryo itself, then the fraction of the abnormally developing embryos should be higher in the *RGTB1*/*rgtb1* progeny, where both *rgtb1*/*rgtb1* and *RGTB1*/*rgtb1* embryos existed.

All the deformed embryos developing after the *rgtb1*/*rgtb1 ×* WT cross showed decreased auxin maxima in the emerging cotyledons and defects in PIN1 localization, resembling mutants in *PIN1* and its regulator *PINOID* [58]. The site of auxin synthesis is normally the domain between the emerging cotyledons, from which it is transported to the cotyledon tips by the PIN1 transporter (reviewed in [59]. The auxin trapped in the domain between the cotyledons promotes growth locally, while its decrease in the cotyledon tips apparently prevents their elongation in embryos developing on the *rgtb1*/*rgtb1* plant. Hence, it may be the absence of auxin in the prospective cotyledon tips and its presence in the prospective apical meristem that leads to extensive embryo growth in the upper domain and reduced growth of the cotyledons [58]. This situation resembles the case where the absence of auxin is important for auxillary meristem formation in the mature sporophyte [81]. In many embryos where defects were observed in cotyledon growth, PIN1 was internalized to intracellular vesicles or remained trapped on the transport route and was unable to reach its final destination on the plasma membrane. Nevertheless, the PIN1 effect cannot explain the maternal influence of the *rgtb1* mutation on embryo development.

To achieve a maternal effect, a molecule must be exuded from the sporophytic tissues surrounding the developing embryo and endosperm, for example from the integuments that form a direct contact with the post-fertilization structures. This molecule may be auxin itself, and indeed, there are reports of auxin flux from the integuments inside the mature ovule around fertilization [79]. If this is the case, an effect of this surplus auxin would most likely be evident at very early stages of the embryo development, which is not observed in case of the *rgtb1* mutation. Another possibility is that a regulatory molecule that acts on auxin levels in the embryo indirectly is produced in the maternal tissues and exported to the embryo sac. Very few genes have been shown to affect the embryo development in a sporophytic maternal mode, but among them are genes encoding for miRNAs [82,83,84,85,86]. For example, the *miRNA167A* gene affects embryo development in a maternal sporophyte-dependent manner [86] and regulates the level of mRNAs for auxin-dependent transcription factors ARF6 and ARF8 [86,87,88]. Levels of miRNA167 in the cell are down-regulated by the absence of auxin which leads to an increase in ARF8 protein [87,88]. The *GH3-2* gene, coding for an enzyme responsible for auxin conjugation to amino acids [89] and silencing of auxin-dependent responses, is positively regulated by ARF8 in rice [87]. Another miRNA that down-regulates the *GH* family of genes is miRNA160, which acts through ARF17 [90].

How can *rgtb1,* a vesicular transport impeding mutation, theoretically impact miRNA synthesis, transport and/or metabolism? Based on our results and literature, we propose an appealing but highly speculative hypothesis. One possible explanation is that the transport of miRNAs from the integuments to the embryo surroundings through plasmodesmata is affected in the *rgtb1* mutant. Plasmodesmata channels are open in the embryo body at least until the late heart stage [91], while connecting the embryo and the suspensor at least until the differentiation of the hypophysis at the globular stage [92,93]. Indeed, plasmodesmata gating through callose deposition at the pore may be dependent on vesicular transport of callose synthases to the neck of plasmodesmata. Although no particular Rab proteins have been proposed so far to be responsible for this process, candidates are for example, but not exclusively, RabA4 [76] or Rab F2b [94]. If this was the case, then disturbance in the Rab prenylation due to the *rgtb1* mutation in the sporophyte would lead to the deficiency of callose synthase at the neck of the plasmodesmal tunnel, opening plasmodesmata and allowing excessive miRNA leakage into the embryo sac (for example miRNA167 [86]). In turn, this would cause *ARF* mRNA degradation and *arf6 arf8* knockdown-like phenotypes. ARF6 and ARF8 are negative regulators of auxin levels in the cell, and the *arf6 arf8* double mutant shows increased free auxin activity in the cytoplasm [88]. A similar mechanism involving miRNAs may then play a role in the differentiation and development of other structures in *rgtb1* mutants, such as the young ovules [15,82,95] or pollen ([54] and this work). Coincidently, other phenotypic traits in the *rgtb1* are reminiscent of *arf6 arf8* double mutant phenotypes, such as dichotomous branching at the apical meristem [53], pollen tube deformations [54], integument asymmetry [15] and anther indehiscence.

### 3.4. The Imbalance of Proteins Building the RGT Complex May Influence Rab Functions in the Embryo Development

Most genes encoding Rab proteins and all genes encoding Rab Geranylgeranyl Transferase complex components are expressed during embryo development in *Arabidopsis*. This reflects the high demand for vesicular traffic in cells undergoing rapid divisions and growth. Genes encoding members of the RGT complex undergo stage-dependent regulation in the embryo; the *REP* gene transcription mirrors *RGTB1* transcription, and is highest from the pre-globular until early heart stages. The *RGTA1* gene transcription shows a bi-phasic mode, with peaks at pre-globular and transition stages of the embryogenesis. While at the pre-globular stage of embryogenesis the transcription of the *RGTB2* gene is relatively high, expression of the *RGTB1* gene peaks from the globular to late heart stages. The disturbances in Rab prenylation and Rab-dependent vesicular transport in the *rgtb1*/*rgtb1*-formed embryo may depend on the balance of the proteins building the RGT complex and the affinity of the Rabs to REP [96] and the REP-Rab complex to RGTA/B heterodimer [97]. The lower level of expression of the RGTB1 encoding gene in the *rgtb1*/*rgtb1* seed coat at the transition stage, when the major deformations in the *rgtb1*/*rgtb1 ×* WT embryos were observed, may reflect the demand for this particular enzyme subunit at this particular time. It suggests that RGTB1 and RGTB2 affinity to particular Rabs (REP-Rab complexes) is not equal, as has been shown for selected Rab proteins during in vitro biochemical experiments [64]. If the transport from the maternal sporophyte to the embryo is RGTB1-dependent, then its deficiency in the seed-coat (at pre-globular stages of the embryo), when *RGTB1* gene expression should be particularly high, may explain the observed effects on the endosperm and embryo development at the following stages.

### 3.5. Both Pre- and Post-Fertilization Events Limit the Seed Set in rgtb1 Mutants

In summary, many factors are responsible for the efficiency of seed formation in *rgtb1* mutants. Importantly, most of them are attributable to disturbed auxin homeostasis in the sporophytic tissues of the parent plant. Anther development and pollen grain formation, ovule and female gametophyte maturation, ovule targeting by the pollen tube, integument/seed coat growth, endosperm development and last but not least embryogenesis depend on auxin balance and timely creation and removal of auxin maxima. In the conditions of perturbed vesicular traffic, auxin transporters such as PIN proteins are mislocalized from their polar site of action and are often internalized from the plasma membrane to vesicles. Auxin transporter mislocalization provides a plausible explanation for creating auxin-enriched domains at the sites where they should not be present and auxin deficit at the sites where the hormone is necessary for correct sporophyte development. Intriguingly, the auxin action seems to be non-autonomous in male and female gametophytes as well as embryo development. The proposed explanation for many *rgtb1* phenotypes, related to aspects of vesicular traffic other than auxin transporters mislocalization, may be the transport of miRNA through plasmodesmata. This very interesting hypothesis awaits direct experimental proof in further studies.

## 4. Materials and Methods

### 4.1. Plant Material and Growth Conditions

Plant lines used: WT Col-0, *rgtb1-1* (SALK 015871) and *rgtb1-2* (SALK 125416) as in [53], *pPIN1:PIN1-GFP*, NASC number N9362 [98], DII-Venus, NASC number N799173 [57] and *pDR5rev**:*3xVenus-NLS, NASC number N799364 [56]. *rgtb1-1* and *rgtb1-2* crosses to fluorescent marker lines are described in [15]. 

For flower and seed observations and for genetic crosses, plants were grown in soil in long day conditions (16 h light, 8 h darkness). Homozygous plants were chosen by means of their characteristic dwarf phenotype and heterozygous plants by genotyping. Flowers were manually crossed as indicated, after emasculation of the white buds two days before crossing. 

A test for seed germination and seedling growth was performed in long day conditions on 1/2 MS medium plates enriched in 1% sucrose and solidified with 1% agar.

### 4.2. Scanning Electron Microscopy

For SEM observations, flowers were fixed in 3% glutaraldehyde in 25 mM phosphate buffer (pH 7.2) overnight, rinsed once in the phosphate buffer and post-fixed in 1% osmium tetroxide overnight. Samples were rinsed three times with distilled water and dehydrated in a graded ethanol series (50, 70, 96 and 100% for 10 min each). Next, specimens were critical-point dried (dried seeds were spilled directly on microscope tables), coated with a thin gold layer and examined using a LEO 1430VP scanning electron microscope (Carl Zeiss).

### 4.3. Aniline Blue Staining

For observation of pollen tube growth through the pistil transmission tract, pistils of WT or *rgtb1*/*rgtb1* flowers were hand pollinated with WT pollen 48 h after emasculation. 8–16 h post pollination pistils were fixed overnight in an acetic acid: ethanol 1:3 solution, hydrated in an ethanol series 75%, 50% and 25%, rinsed in water and incubated overnight in 1M NaOH at room temperature. After washing in water, the specimens were stained overnight in a decolorized aniline blue solution [99]. Pistils were observed under an Eclipse E-800 epifluorescence microscope (Nikon) with an Epi-Fl Filter Block N UV-2A (EX 330-380, DM 400, BA 435-485) and photographed with an ORCA ER monochromatic camera (Hammamatsu) or Nikon DS-5Mc CCD camera (PRECOPTIC Co.).

### 4.4. Ruthenium Red Staining

Oligosaccharide halo formation during seed germination was observed by ruthenium red staining of the pectins formed around seeds. After 20 min of imbibition seeds were treated with a ruthenium red solution (0.25 mg/mL) and incubated for 15 min at room temperature [100]. Seeds were observed under an Eclipse E-800 epifluorescence microscope (Nikon) and photographed with an ORCA ER monochromatic camera (Hammamatsu) or were observed under an Olympus SZ61 stereoscope and photographed with a Canon 750D camera.

### 4.5. Alexander Staining of Pollinated Stigma

Flowers were emasculated 48 h before pollination with appropriate pollen. After 4 h, the pistils were removed and fixed in Carnoy solution overnight and stained with Alexander stain according to [101]. Stigmas were observed under an Olympus SZ61 stereoscope and photographed with a Canon 750D camera.

### 4.6. Semi-Thin Sections Preparation 

Semi-thin sections for anthers analysis were prepared using the standard method for TEM as previously described [102].

### 4.7. Sample Clearing

Siliques were fixed in acetic acid: ethanol 1:3 solution and cleared in chloral hydrate solution (66.7% chloral hydrate (*w*/*w*), 8.3% glycerol (*w*/*w*)) or cedar oil as described [102]. Ovules or embryos were examined under a Nikon Eclipse E800 epifluorescence microscope equipped with differential interference contrast (DIC) optics and a Nikon DS-5Mc CCD camera (PRECOPTIC Co.).

### 4.8. Fluorescence Analysis of Ovules 

For fluorescence analysis, the ovules or manually isolated embryos were mounted in 7% glucose. For FM^®^ 4–64 dye application (Invitrogen), dissected ovules or embryos were incubated in 4 µM FM^®^ 4–64 diluted in 7% glucose, on glass slides for 1–2 h in the dark and observed using a Confocal Laser Scanning Microscope (CLSM; Leica TCS SP8X). To avoid differences in fluorescence intensity due to transcript silencing in consecutive generations, sister (progeny of the same mother plant) WT and *rgtb1* plants were used for imaging. Specimens were imaged using an epifluorescence microscope (Nikon Eclipse E800) or CLSM. Detection of GFP and YFP (VENUS) under the epifluorescence microscope was achieved with Epi-Fl Filter Block B-1E (EX 470–490, DM 505, BA 520–560). A filter Block G-2A (EX 510–560, DM 575, BA 590) was used for co-localization of non-specific fluorescent signal and cuticle-like components. 

Under CLSM, the GFP excitation wavelength was set to 489 nm and emission was detected at 505–547 nm; for YFP (VENUS)—ex 514 nm, em 524–566 nm. For FM^®^ 4–64—ex 558 nm, em 674–766 nm. CLSM imaging was performed with 63× oil immersion objective.

All figures were prepared in Adobe Photoshop (Elements 11 and CS6 versions).

### 4.9. Statistical Analysis

All calculations were performed in GraphPad Prism 5.0 software. Mean values and standard deviation (SD) were calculated and data were compared with a two-tailed unpaired Student t-test. In cases where proportions were compared, the z-test for two proportions was used, assuming large samples (using normal distribution). In case of germination efficiency, the χ^2^ test was used, comparing the observed genotype segregation and the theoretical Mendelian inheritance. Graphs were prepared using the same software. To avoid possibility that the environmental conditions in the greenhouse influence the studied phenomena at least three independent plant cultivations, each with at least 10 plants per genotype, were performed to generate data. The number of observed structures (ovules/seeds/pollen grains, embryos, etc.) is given in the graphs. 

### 4.10. Transcriptomic Analysis

*In silico* expression profiles were extracted from [63,65]. The dataset in [63] incorporates multiple stages of embryo development including 2-cell, 8-cell, 32-cell, preglobular, globular, early heart, late heart, early torpedo, late torpedo, bent cotyledon and mature green embryos. RNAseq data were downloaded from the NCBI Gene Expression Omnibus (GEO; https://www.ncbi.nlm.nih.gov/geo/, accessed on 29 June 2021) under accession number GSE121236. Data from [65] were obtained from http://seedgenenetwork.net/ (accessed on 29 June 2021), which incorporates microarray profiles of laser dissected *Arabidopsis* seed compartments at different stages of development. The stages of development include: pre-globular, globular, heart, linear cotyledon and green cotyledon; the tissues include: chalazal endosperm, chalazal seed coat, embryo proper, general seed coat, micropylar endosperm, peripheral endosperm and whole seed.

## 5. Conclusions

Mutants in the *Arabidopsis* Rab Geranylgeranyl Transferase β1 subunit-encoding gene have lower seed set due to defects in pre- and postfertilization stages. Pollen reception at the stigma and pollen tube guidance are decreased. Mature *rgtb1* ovules form autonomous endosperm at a high frequency due to increased auxin levels inside the embryo sac. *rgtb1* plants exert a sporophytic maternal effect on the embryos at the transition stage by releasing an unknown factor from the seed coat acting non-autonomously. At the preceding stages of development, *RGTB1* expression in the seed coat is high in the WT plants, while it is absent in *rgtb1* plants. These observations highlight the importance of Rab-dependent vesicular traffic in seed formation in angiosperms.

## Figures and Tables

**Figure 1 ijms-22-07907-f001:**
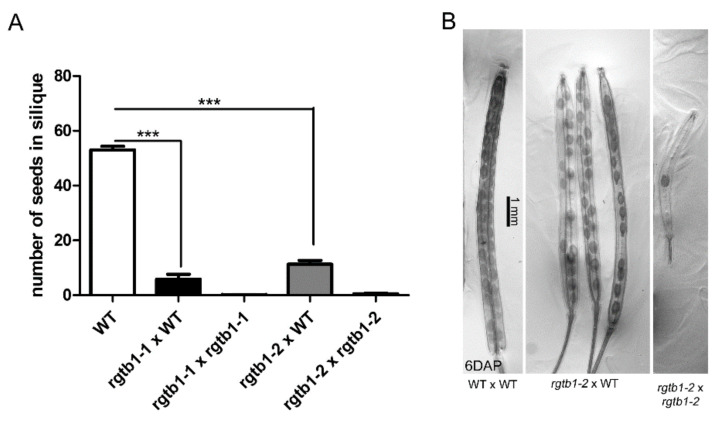
Number of seeds per silique is decreased in *rgtb1*/*rgtb1* pistils that were either left for open pollination or manually pollinated with WT pollen. (**A**) Graph showing the number of seeds per silique. Number of siliques counted: WT *n* = 21, *rgtb1-1 ×* WT *n* = 19, *rgtb1-1 × rgtb1-1 n* = 12, *rgtb1-2 ×* WT *n* = 34, *rgtb1-2 × rgtb1-2 n* = 19. Bars show mean ± SD. Values were compared with a two-sided Student *t*-test; *** denotes a *p* value < 0.001. (**B**) Siliques 6 days after hand pollination (6DAP) showed differences: WT silique filled with seeds, an intermediate situation in the *rgtb1 ×* WT cross and a silique from a self-pollinated *rgtb1* plant with only one seed. Light microscopy, representative images are shown. Scale bar represents 1 mm, common to all images.

**Figure 2 ijms-22-07907-f002:**
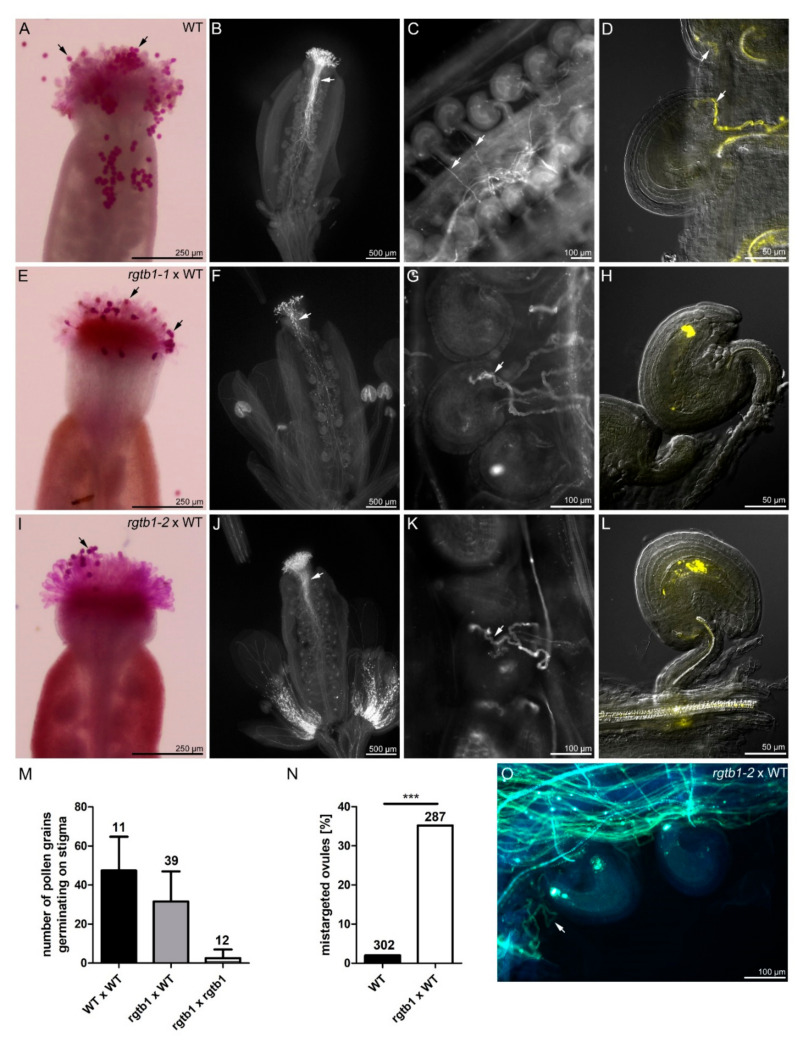
Pollen tube germination, growth and guidance are compromised in *rgtb1* plants. Reception of pollen on stigma and growth of pollen tubes through the transmission tract and toward ovules in WT (**A**–**D**) and *rgtb1* mutants (**E**–**L**,**O**). (**A**,**E**,**I**) Alexander-stained (purple) pollen on WT (**A**), *rgtb1-1* (**E**) and *rgtb1-2* (**I**) stigmas, 4 h after hand pollination (4HAP) with WT pollen (black arrows). Most mutant grains hydrated and germinated, although the capacity of the *rgtb1* stigma to attach WT pollen was decreased in comparison to WT stigma reception (**E**,**I**). Scale bar 250 μm, common to images (**A**,**E**,**I**). Quantification of pollen grains germinating on stigmas, bars show mean ± SD of pollen grains germinating per stigma; number above bars describe number of stigmas counted per experimental cross. Genotypes used for each cross are given in the bar description; pollen donor is given first and pollen acceptor second. (**B**–**D**,**F**–**H**,**J**–**L**,**O**) Pistils 8 h after hand pollination (8HAP) with WT pollen were stained with decolorized aniline blue to visualize callose. (**B**,**F**,**J**) Bundles of pollen tubes growing through the transmission tract; white arrows point to the WT pollen tubes growing in the WT transmission tract (**B**) or *rgtb1* transmission tracts (**F**,**J**), scale bar 500 μm, common to images (**B**,**F**,**J**). (**C**,**D**,**G**,**H**,**K**,**L**,**O**) Examples of pollen tubes growing toward the micropyle. (**C**,**D**) Normal WT pollen tube guidance in WT pistil. (**G**,**K**,**O**) More than one WT pollen tube twisting around the micropyle (white arrow) in *rgtb1* pistils. (**H**,**L**) No WT pollen tubes approaching *rgtb1* ovules of apparently normal size and integument shape, but showing a callose plug at the site of the egg apparatus. Scale bar 100 μm in (**C**,**G**,**K**,**O**) and 50 μm in (**D**,**H**,**L**,**N**), quantification of incorrect WT pollen tube guidance to the micropyle in *rgtb1* mutant transmission tract. (**M**) Quantification of pollen grains germinating on stigmas. Pollen grains forming pollen tubes were counted on stigmas stained with DAB (decolorized aniline blue) 18 h after hand pollination (18HAP). Bars show mean ± SD of pollen tubes germinating on a stigma. Number of pistils observed is given above the bars. Pollen acceptor is given as first, pollen donor as second in the graph legend. Numbers above the bars describe the number of ovules counted per genotype. Values were compared with a *z*-test for two proportions; *** denotes a *p* value < 0.001.

**Figure 3 ijms-22-07907-f003:**
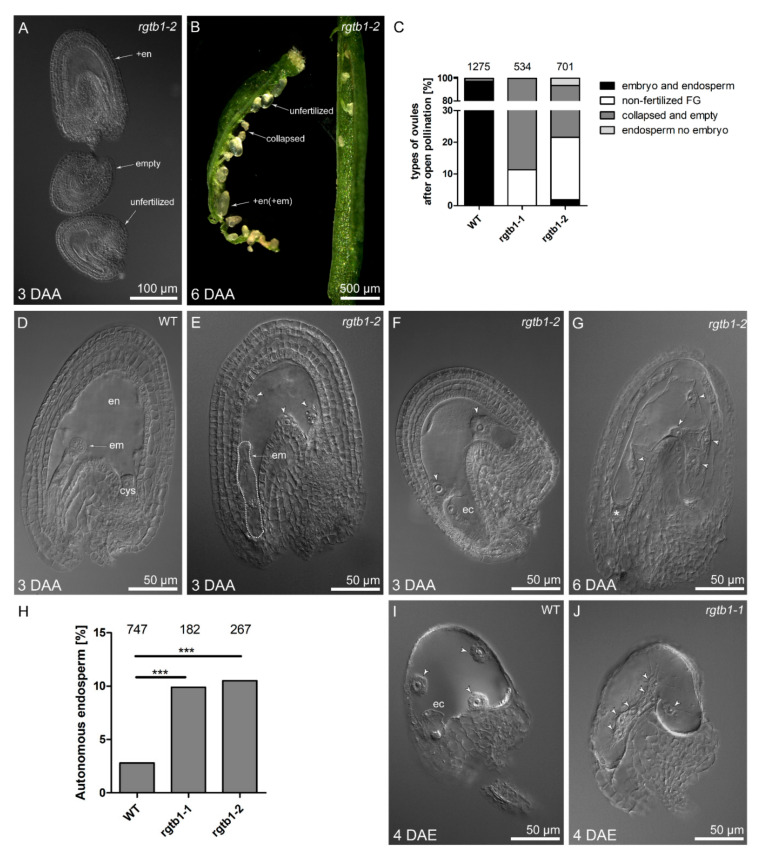
Post-anthesis stages in *rgtb1*. Types of ovules after pollination (**A**–**G**) or emasculation (**H**–**J**). (**A**,**B**) Several types of neighboring ovules were noted in the same silique of *rgtb1* mutant after self-crossing. White arrows point to shrunken unfertilized empty ovules (not containing FG), mature unfertilized ovules and enlarged ovules after endosperm (and embryo) initiation. (**A**) Cleared ovules observed in DIC contrast, scale bar 100 μm; (**B**) representative example of one fresh silique dissected in two parts observed under a stereomicroscope, scale bar 500 μm. (**C**) quantification of the types of ovules found in WT vs. *rgtb1* mutants after self-pollination: ovule with embryo and endosperm that were dominating in WT (**D**), but noticed very rarely in *rgtb1* (**E**) and seed-like structures with enlarged central cell or endosperm found only in *rgtb1* (**F**,**G**). Scale bar 50 μm for images (**D**–**G**). The fraction of ovule types after open pollination in WT and *rgtb1* mutants was counted in cleared preparations under a microscope 6 days after open pollination (6DAP). The number of ovules counted for each genotype is given above the bars. (**H**–**J**) Ovules in WT and *rgtb1* after emasculation form autonomous endosperm. (**H**) Quantification of ovules forming autonomous endosperm (AE) in WT and *rgtb1* mutants. AE formation was counted in cleared preparations or in freshly isolated ovules under a microscope 4 days after emasculation (4DAE). The total number of ovules counted for each genotype is given above each bar. Values were compared with a z-test for two proportions; *** denotes a *p* value < 0.001. (**I**,**J**) Examples of AE in both WT (**I**) and *rgtb1* (**J**) emasculated flowers. AE have several nuclei (**I**,**J**, arrowheads). Scale bar 50 μm for images (**I**,**J**). Abbreviations: ec—egg cell; cys—chalazal cyst, en—endosperm, em—embryo, DAA—day after anthesis, DAE—day after emasculation. Dotted line highlights the cells of the embryo.

**Figure 4 ijms-22-07907-f004:**
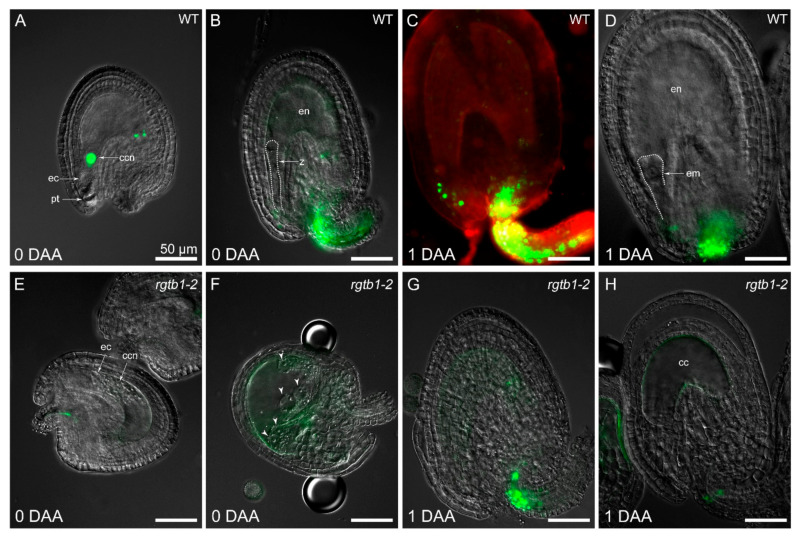
Auxin signaling is different in WT and *rgtb1* ovules after open pollination. (**A**) signal of auxin reporter *pDR5rev*:3xVenus-NLS (green) is present inside the embryo sac, but not in the sporophytic tissues surrounding it in WT ovules. (**B**–**D**) Shortly after fertilization in WT plants, the *pDR5rev:*3xVenus-NLS signal appears in the funiculus and chalaza, but is no longer present in the developing endosperm or embryo. (**E**–**H**) in *rgtb1* ovules, the *pDR5rev:*3xVenus*-*NLS signal is absent or very weak in the embryo sac and the sporophytic tissues of the ovule before (**E**) and after anthesis (**F**–**H**), although the formation of the autonomous endosperm is sometimes obvious (**F**); white arrowheads point to the AE nuclei. (**C**) CSLM image, the *pDR5rev*:3xVenus-NLS signal (green) are counterstained with FM4-64 (red) to contour the cells, all the other images were made using fluorescence microscopy, showing the *pDR5rev*:3xVenus-NLS signal (green). The scale bar of 50 μm in (**A**) is the same in all images. Abbreviations: cc—central cell, ccn—central cell nucleus, ec—egg cell, pt—pollen tube, en—endosperm, z—zygote, em—embryo, DAA—day after anthesis.

**Figure 5 ijms-22-07907-f005:**
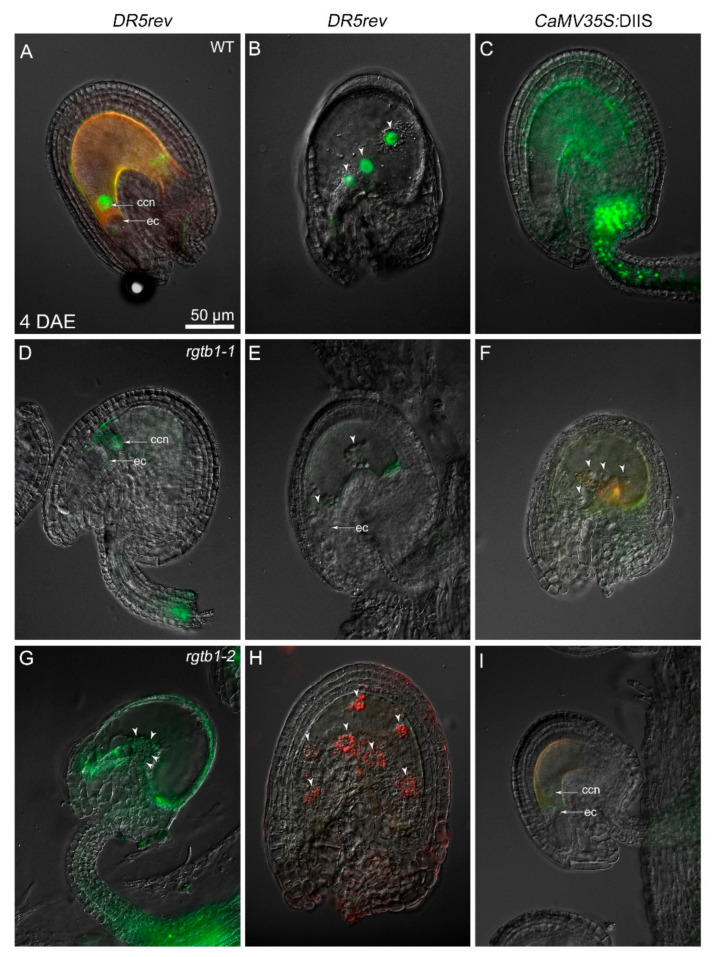
*rgtb1* ovules resemble ovules in emasculated WT plants**.** Ovules 4 days after manual emasculation (4DAE) of flowers. (**A**,**B**) Auxin reporter *pDR5rev*:3xVenus-NLS signal, marking canonical auxin-dependent transcription response, is present in the central cell nucleus of unfertilized WT flowers (**A**) and autonomous endosperm nuclei formed in ovules of emasculated WT flowers (**B**). (**C**) DIIS-Venus auxin reporter (green) is present in the integuments of unfertilized ovules, marking an absence of auxin. (**D**,**E**,**G**,**H**) *pDR5rev*:3xVenus-NLS signal is absent or weak in the central cell nucleus (**D**) of the unfertilized *rgtb1* ovules or AE nuclei (**E**,**G**,**H**), shown by the white arrowheads, marking an absence of auxin transcription response. (**F**,**I**) DIIS-Venus signal is absent from the sporophytic tissues of the unfertilized *rgtb1* ovules, both forming AE nuclei (**F**), as shown by the white arrowheads), and non-forming AE (**I**), marking the presence of auxin in them. Fluorescence microscopy—all images; *pDR5rev*:3xVenus-NLS (green) or DIIS-Venus (green) as indicated, unspecific fluorescence (red). The scale bar of 50 μm in (**A**) is the same for all images. Abbreviations: ccn—central cell nucleus, ec—egg cell, DAE—days after emasculation.

**Figure 6 ijms-22-07907-f006:**
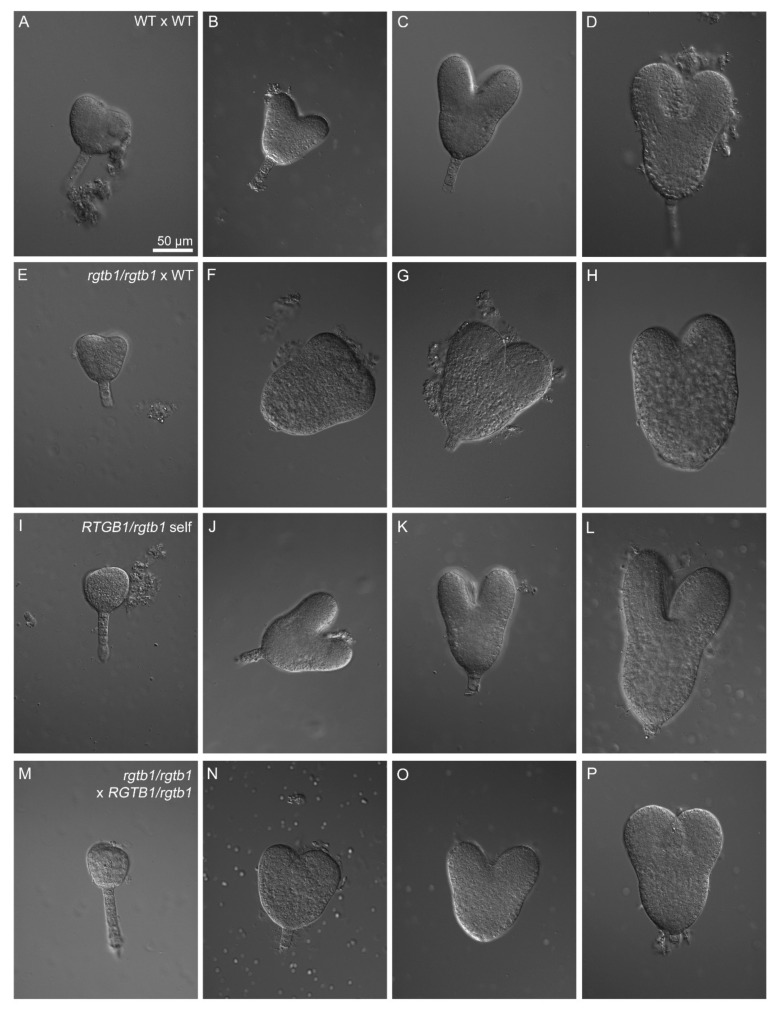
Deformed embryos form on the *rgtb1* mother plant regardless of the pollen donor. Morphology of the embryos from the early heart stage until the torpedo stage formed on WT or *rgtb1* mother plants; microscopy images in DIC contrast. (**A**–**D**) Embryos formed on the WT plant after self-pollination, (**E**–**H**) embryos formed on the *rgtb1* plant after pollination with WT pollen, (**I**–**L**) embryos formed on the *RGTB1*/*rgtb1* plant after self-pollination and (**M**–**P**) embryos formed on the *rgtb1* plant after pollination with pollen coming from *RGTB1*/*rgtb1* plant. Note the characteristic widening in the middle part of the embryo and cotyledon shortening in images (**F**–**H**) and (**N**–**P**); no changes were seen in the suspensor and hypophysis formation in all images. The scale bar of 50 μm in (**A**) is the same in all images.

**Figure 7 ijms-22-07907-f007:**
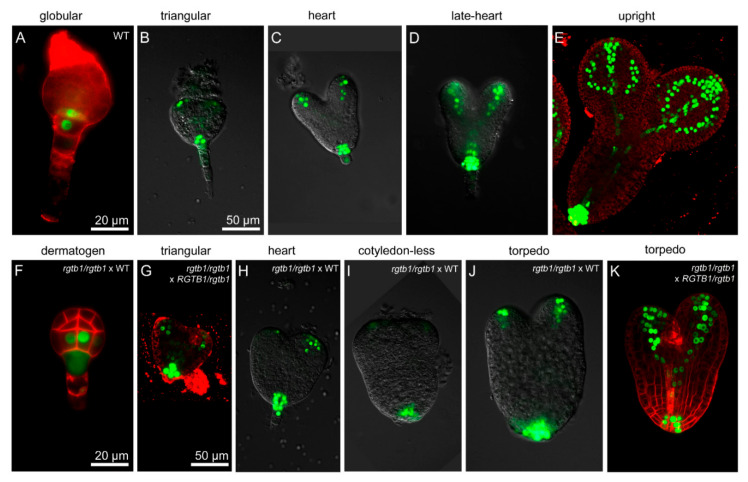
Deformed embryos in *rgtb1* plants show diffuse auxin maxima. Auxin maxima monitored by the expression of the *pDR5rev*:3xVenus-NLS reporter (green) in embryos formed on WT or *rgtb1* plants. Auxin is abundant in a basal half of the globular embryos (**A**,**F**). Strong auxin maxima form in the prospective cotyledon tips and prospective root meristem at the triangular stage of the embryos (**B**,**G**). Starting from the early heart stage auxin maxima at the tips of the cotyledons become more diffuse and weaker in embryos formed on *rgtb1* plants (**H**–**J**) compared to embryos formed on WT plants (**C**,**D**). Auxin maximum in the embryo basal part remains strong, regardless of the genotype of the mother plant (**B**–**D**,**G**–**J**). Note the larger size, wider middle part of the embryo and shorter cotyledons in the *rgtb1*-formed embryos (**H**–**J**). *rgtb1*-formed embryos are delayed in cotyledon vasculature formation (**J**,**K**) in comparison to WT embryos (**D**,**E**). (**E**–**G**,**K**) CSLM images, *pDR5rev*:3xVenus-NLS (green) counterstained with FM4-64 (red) to visualize cell contours. (**A**–**D**,**H**–**J**) Fluorescence microscopy, *pDR5rev*:3xVenus-NLS (green). The scale bar is 20 μm for images (**A**,**F**) and 50 μm for all the other images.

**Figure 8 ijms-22-07907-f008:**
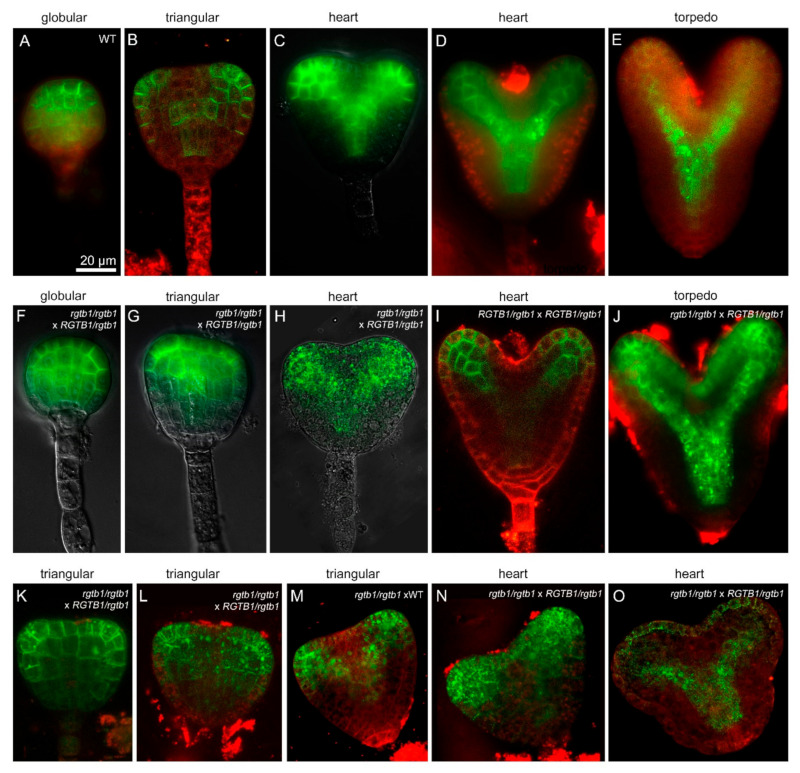
Deformed embryos growing on *rgtb1* plants show PIN1 protein mislocalization*. pPIN1*:PIN1-GFP expression in embryos in WT (**A**–**E**) and *rgtb1* plants (**F**–**O**). PIN1-GFP protein is often mislocalized from the plasma membrane to intracellular vesicles in embryos formed on *rgtb1* plants (**H**,**J**–**O**) in comparison to WT embryos (**A**–**E**). Embryos formed on *rgtb1* plants at transition and heart stages of development show a wider domain of *pPIN1*:PIN1-GFP expression in the prospective apical meristem (**G**–**H**,**K**–**N**) than WT embryos at corresponding stages (**B**,**C**). (**B**,**I**,**K**–**O**) CLSM images, *pPIN1*:PIN1-GFP signal (green) counterstained with FM4-64 (red) to visualize cell contours; (**A**,**C**–**H**,**J**) fluorescence microscopy, *pPIN1*:PIN1-GFP signal (green). The scale bar of 20 μm in (**A**) is the same in all images.

**Figure 9 ijms-22-07907-f009:**
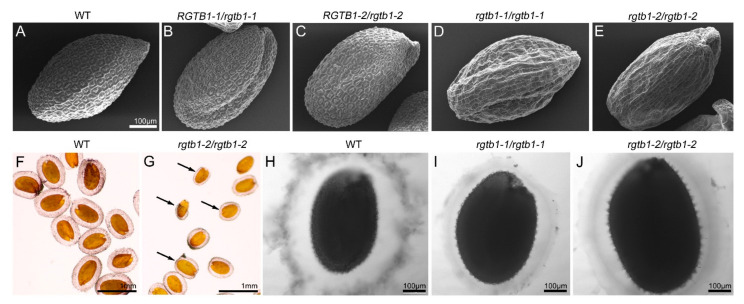
Seed coat formation is affected in *rgtb1* plants. The seed coat of WT seeds (**A**) and seeds formed on *RGTB1*/*rgtb1* self-crossed plants (**B,C**) appear similar. However, *rgtb1*-derived seed coats consist of cells of uneven shape that do not contain mucilage pockets (**D**–**E**). SEM images, scale bar 100 μm, correspond to (**A**–**E**). After imbibition, seeds formed on WT plants exude large amounts of a mucilage sheath (**F**,**H**), but *rgtb1*-derived seeds form a more compact and uneven mucilage sheath (**G**,**I**,**J**). Scale bar 1 mm for images (**F**,**G**); scale bar 100 μm for images (**H**–**J**); (**F**–**J**) microscopic images of seeds stained with ruthenium red. Black arrows in (**G**) point to the seeds forming uneven or no mucilage sheath.

**Figure 10 ijms-22-07907-f010:**
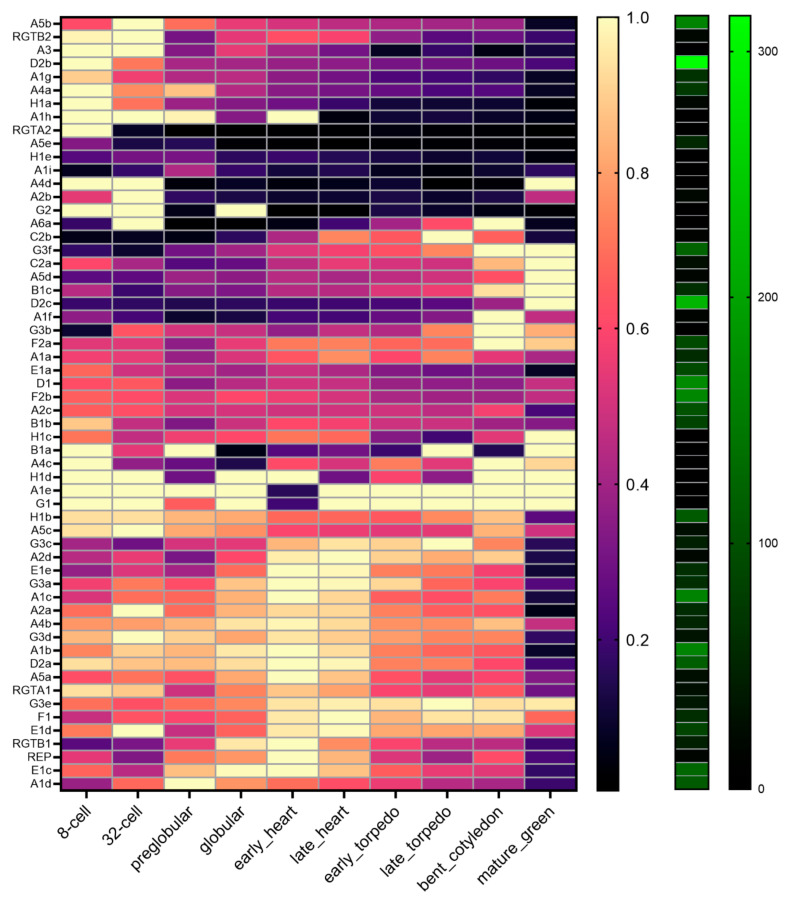
Transcriptomic analysis of RGT and Rab components during embryo development. In silico expression profiles during embryogenesis of the Rab and RGT complex genes were examined using publicly available transcriptome data [63]. RNAseq data were downloaded from the NCBI Gene Expression Omnibus (GEO; https://www.ncbi.nlm.nih.gov/geo/, accessed on 29 June 2021) under accession number GSE121236. Average expression values for each gene at each stage were calculated and then presented as a proportion of the maximum expression value for each gene, providing a relative expression value between 0 and 1. Expression profiles were clustered using hierarchical clustering (Pearson correlation) and heat maps were generated in GraphPad Prism 9.0.0. The maximum RPKM expression value for each gene in any given stage is indicated in the black-to-green heatmap on the right.

## Data Availability

The data presented in this study are openly available in http://eprints.ibb.waw.pl/ (accessed on 29 June 2021) repository.

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
