# Peer review of "The Rab Geranylgeranyl Transferase Beta Subunit Is Essential for Embryo and Seed Development in *Arabidopsis thaliana"

_ijms, 2021, doi:10.3390/ijms22157907_

Round 1
Reviewer 1 Report
Rojek et al., reported in this manuscript that RTGB is essential for embryo and seed development in Arabidopsis. The manuscript is written well, most figures are at high quality. Although at least three papers about RTGB have already been published, the observations from this study seem new. The introduction is quite comprehensive. I suggest the authors to provide some more information about the novelty and new findings from this manuscript in comparison to previous published 3 papers at the end of the Introduction or somewhere in the end of Results or Discussion. What is lacking from previous studies?
Please find below some of my other comments:
- Lines 57, 129, 258, 316, 546, 555, 564, delete extra space.
- Line 121, to me, it is just partially not fully recovered, right? Please revise to be more precise.
- Figure 3H, please provide error bars.
- Line 340, delete extra “it” after PIN1.
- Lines 345-348, there is something wrong here. Please correct. How it is possible that only Fig. 8M is rgtb1-/-WT, while all the results from lines 345-350 are talking about rgtb1/rgtb1 x WT? Please revise to make it clear and be consistent for the labelling in different figures throughout the manuscript.
- Line 393, delete the extra “,”.
- Figure 10 and Figure S3, all genes should be italics.
Overall, I suggest that minor revisions are needed.
Author Response
Reviewer: Rojek et al., reported in this manuscript that RTGB is essential for embryo and seed development in Arabidopsis. The manuscript is written well, most figures are at high quality. Although at least three papers about RTGB have already been published, the observations from this study seem new. The introduction is quite comprehensive. I suggest the authors to provide some more information about the novelty and new findings from this manuscript in comparison to previous published 3 papers at the end of the Introduction or somewhere in the end of Results or Discussion. What is lacking from previous studies?
Authors: We improved the paragraph in the Introduction describing what is known about the consequences of decreasing the RGT activity in plants. We stated directly what was described in the previous studies (influence of rgtb1 mutation on the mature sporophyte, pollen and female gametophyte development) and what remained unknown (influence of this mutation on the seed and embryo development).
Lines 57, 129, 258, 316, 546, 555, 564, delete extra space (and other spacing mistakes).
Authors: We corrected all spacing mistakes that we could find.
Line 121, to me, it is just partially not fully recovered, right? Please revise to be more precise.
Authors: We added remarks on the penetrance of the rgtb1 and rgtb2 mutations in Arabidopsis.
Figure 3H, please provide error bars.
Authors: Figure 3H (and Fig. 2N) show proportion of AE formation (ovule mistargeting) in all observed ovules formed in rgtb1/rgtb1 plants. The material was gathered from a few independent experiments (each consisting of about 10 plants) performed at different times. The graphs show proportion of pulled observations from all the experiments and were therefore analyzed with the z-test for two-proportions. Numbers of observations were high so we used this test assuming normal distribution.
Lines 345-348, there is something wrong here. Please correct. How it is possible that only Fig. 8M is rgtb1-/-WT, while all the results from lines 345-350 are talking about rgtb1/rgtb1 x WT? Please revise to make it clear and be consistent for the labelling in different figures throughout the manuscript.
Authors: We made the labelling consistent and decided to use the rgtb1/rgtb1 notation for the homozygote and RGTB1/rgtb1 notation for the heterozygote. We only preserved WT notation for RGTB1/RGTB1 genotype, but explained it early in the text, at the beginning of the Results section.
Figure 10 and Figure S3, all genes should be italics.
Authors: We agree with the reviewer, but we left the non-italics names for the sake of better resolution of the (small) font on the figures 10 and S3. We italicised all the genes names in the text.
Reviewer 2 Report
The reviewed manuscript contains the results of a high-quality investigation combining embryology, physiology, genetics and transcriptomics. There are several minor corrections and suggestions marked directly in the manuscript file (see attached). The only more or less serious methodological question refers to statistical analysis, as t-test has its limitations when working with small samples. As far as I have understood from the text, samples were small. If not, this should be additionally emphasized.
I also suggest to list briefly the main conclusions at the very end of a text, as they are somewhat 'lost' in discussion.
After considering the suggestions and corrections, this paper can be published in the IJMS.

Author Response
Reviewer: The reviewed manuscript contains the results of a high-quality investigation combining embryology, physiology, genetics and transcriptomics. There are several minor corrections and suggestions marked directly in the manuscript file (see attached).
Authors: We introduced all the suggested corrections (spelling, italics, grammar etc.).
The only more or less serious methodological question refers to statistical analysis, as t-test has its limitations when working with small samples. As far as I have understood from the text, samples were small. If not, this should be additionally emphasized.
Authors: We used three types of statistical test in this work: 1) in case of absolute numbers (number of pollen grains attached per stigma, number of seeds per silique, etc.) we showed mean +/- SD on graphs and analyzed the data with Student t-test. The analyzed samples were large and the data distribution normal (statistical analysis performed in GraphPad). The exact numbers of observations are given above the bars on graphs or in figure descriptions, 2) in case of proportion comparisons (proportion of mistargeted ovules among all observed, proportion of deformed embryos among all observed, etc.) we showed proportion of particular observations in the pull of all the observations performed for a given genotype and analyzed the data with z-test for two-proportions. The numbers of observations were large so we assumed normal distribution. The exact numbers of observations are given above the bars on graphs, 3) in case of the analysis of the genotype distribution in comparison to the theoretical Mendelian inheritance we used the χ2 test.
I also suggest to list briefly the main conclusions at the very end of a text, as they are somewhat 'lost' in discussion.
Authors: We added the conclusions paragraph at the end of discussion to summarize our main findings.